# Spatial decoupling of bromide-mediated process boosts propylene oxide electrosynthesis

Mingfang Chi[1,6], Jingwen Ke[1,6], Yan Liu[1,6], Miaojin Wei[1], Hongliang Li [1], Jiankang Zhao[1], Yuxuan Zhou[1], Zhenhua Gu [1], Zhigang Geng [1] ✉ & Jie Zeng [1,2,3,4,5] ✉

The electrochemical synthesis of propylene oxide is far from practical application due to the limited performance (including activity, stability, and selectivity). In this work, we spatially decouple the bromide-mediated process to avoid direct contact between the anode and propylene, where bromine is generated at the anode and then transferred into an independent reactor to react with propylene. This strategy effectively prevents the side reactions and eliminates the interference to stability caused by massive alkene input and vigorously stirred electrolytes. As expected, the selectivity for propylene oxide reaches above 99.9% with a remarkable Faradaic efficiency of 91% and stability of 750·h (>30 days). When the electrode area is scaled up to 25 cm$^2$, 262 g of pure propylene oxide is obtained after 50·h continuous electrolysis at 6.25 A. These findings demonstrate that the electrochemical bromohydrin route represents a viable alternative for the manufacture of epoxides.

Propylene oxide (PO) is a versatile feedstock in the production of various chemicals including polyether polyol, propylene glycol, and dimethyl carbonate[1–4]. The global consumption of PO exceeded 10 million tonnes in 2021, accelerating with an annual growth rate of 6%[5–7]. Currently, the industrial manufacture of PO mainly relies on the chlorohydrin and hydrogen peroxide to PO (HPPO) processes[8–10]. The chlorohydrin process has been gradually obsoleted since it produces a large amount of effluent and sludge[11,12]. Although the hydrogen peroxide-based process is environmentally benign, it is restricted by the high cost of H$_2$O$_2$[11,13]. An alternative, environmentally sustainable method to produce PO is through the electrochemical pathway, with electricity from renewable solar or wind energies[14,15].

Recently, the electrochemical synthesis of PO has attracted substantial attention[11,16]. The direct electrochemical synthesis of PO from propylene is far from practical application due to the limited Faradaic efficiency (FE) (<20%), current density (<5 mA cm$^{-2}$), and stability (<10 h). Sargent et al. put forward a chloride-mediated system for the selective electrochemical synthesis of PO, which achieved a current density of 1 A cm$^{-2}$, FE of ~70%, and product selectivity of ~97%[17,18]. The current density and selectivity were significantly enhanced, whereas the unsatisfied FE still restricted its practical application. The major loss of FE is ascribed to the irreversible cleavage of hypochlorous acid (HOCl) to unreactive ClO$^-$ [19]. In addition, the fierce competition of the undesirable oxygen evolution reaction (OER) also limited the FE since the standard electrode potential of Cl$^-$ to Cl$_2$ (1.36 V versus reversible hydrogen electrode, vs RHE) is higher than that of OER (1.23 V vs RHE)[20,21]. Br$^-$/Br$_2$ redox mediators were developed for the selective oxidation of alkenes to circumvent the shortcomings of the electrochemical chlorohydrin route[22–26]. However, their performance (including activity, stability, and selectivity) still suffers from the

[1]Hefei National Research Center for Physical Sciences at the Microscale, University of Science and Technology of China, 230026 Hefei, Anhui, P. R. China. [2]CAS Key Laboratory of Strongly-Coupled Quantum Matter Physics, University of Science and Technology of China, 230026 Hefei, Anhui, P. R. China. [3]Key Laboratory of Surface and Interface Chemistry and Energy Catalysis of Anhui Higher Education Institutes, University of Science and Technology of China, 230026 Hefei, Anhui, P. R. China. [4]Department of Chemical Physics, University of Science and Technology of China, 230026 Hefei, Anhui, P. R. China. [5]School of Chemistry & Chemical Engineering, Anhui University of Technology, 243002 Ma'anshan, Anhui, P. R. China. [6]These authors contributed equally: Mingfang Chi, Jingwen Ke, Yan Liu. ✉e-mail: gengzg@ustc.edu.cn; zengj@ustc.edu.cn

following challenges. Firstly, the operation current was limited (<0.35 A). Specifically, the alkenes were bubbled or placed into the electrolyzer directly in these works. In this case, the alkenes inevitably underwent unwanted side reactions at high currents such as over-oxidation, which was not conducive to the PO production at high currents, thus restricting the scalable production of PO[11,16,27]. At the same time, the side reactions caused the loss of alkenes, resulting in the rising cost of industrial manufacture. Secondly, the current stability was impeded by the massive input of alkene and vigorously stirred electrolytes. Thirdly, the FE was limited since the generated HBrO was proved to suffer from cleavage reactions when exposed to metal-based anodes with specific adsorption sites. Fourthly, halogenated products might be produced as the local concentration of $Br_2$ at the anode surface was relatively high. The generation of brominated products restricted the reaction selectivity, leading to higher product separation costs.

Herein, we spatially decoupled the electrolysis process and the propylene conversion process by utilizing the bromide mediator. In detail, the bromine was generated at the anode and then transferred into an independent reactor to react with propylene. The spatially decoupled system avoided direct contact between the anode and propylene, which was expected to maintain the activity and selectivity even at high currents. Additionally, benefiting from the decoupled system, the interference of system stability caused by massive feedstock input was eliminated. As expected, the developed electrochemical bromohydrin system enabled the efficient synthesis of PO using cost-effective carbon paper as the anode. At all applied potentials, the selectivity for PO reached above 99.9%. Notably, at 1.9 V vs Ag/AgCl, a remarkable FE of 91% was achieved. When this system ran at 250 mA cm$^{-2}$ for 750 h (>30 days), the FE for PO ($FE_{PO}$) exhibited only 0.73% decay. We also designed an enlarged flow reactor with a geometric electrode area of 25 cm$^2$, combined with a distillation separation device to demonstrate the practical application promise of our strategy. After 50-h continuous electrolysis at 6.25 A, 262 g of pure PO was obtained. In addition, the electrochemical bromohydrin route was also applicable for the efficient transformation of other alkenes, including gas alkenes (such as ethylene, 1-butylene, and isobutylene) and liquid alkenes (such as 1-octene, cyclopentene, and styrene). These results undoubtedly validated the feasibility of the electrochemical bromohydrin route for the synthesis of epoxides.

## Results

### Reaction pathway

Figure 1 illustrates the reaction procedure of the electrochemical bromohydrin route for the synthesis of PO. The Br$^-$ in the electrolyte was first oxidized to $Br_2$ at the anode, which subsequently transformed into HBrO through the disproportionation reaction. The generated HBrO was transferred into an independent reactor, then reacting with propylene to form propylene bromohydrin ($C_3H_7OBr$). Meanwhile, the $H_2O$ was split into hydrogen ($H_2$) and OH$^-$ at the cathode. Finally, PO was produced through the saponification process between $C_3H_7OBr$ and OH$^-$. Moreover, a distillation column was designed to obtain pure PO from the electrolyte. After the separation, the electrolyte was reused for a new round of electrolysis. We conducted in situ Raman measurements to monitor the electrooxidation process from Br$^-$ to $Br_2$ (Supplementary Fig. 1a). Besides, the Raman spectra of the products after addition and saponification processes confirmed the generation of $C_3H_7OBr$ and PO, respectively (Supplementary Fig. 1b, c).

### Catalytic performance for the synthesis of PO via the electrochemical bromohydrin route

In a proof-of-concept experiment, we evaluated the catalytic performance for the synthesis of PO via the electrochemical bromohydrin route in an H-cell with potassium bromide (KBr) as the electrolyte. We initially investigated the performance of various commercial electrodes via linear sweep voltammogram (LSV) measurements. The carbon paper exhibited a higher current density than other commercial electrodes, implying the superior catalytic performance of the carbon paper (Supplementary Fig. 2). Besides, as the benchmark catalysts for the conventional chlor-alkali process, carbon-based materials with corrosion-resistance, cost-effectiveness, and high electrical conductivity are regarded as promising candidates for the electrochemical bromohydrin route[28–30]. As such, carbon paper composed of highly graphitic fibers served as the anode (Supplementary Fig. 3). Tafel and electrochemical reaction order analyses demonstrated that the Br$^-$ electrooxidaiton over the carbon paper was operated via the Volmer-Heyrovsky mechanism with the Heyrovsky step as the rate-determining step (Supplementary Fig. 4). The gaseous and liquid products during the reaction were confirmed and quantified via online gas chromatography (GC) and $^1$H nuclear magnetic resonance ($^1$H NMR) spectroscopy, respectively (Supplementary Figs. 5, 6). Given that the oxidation of Br$^-$ was the first step of the electrochemical

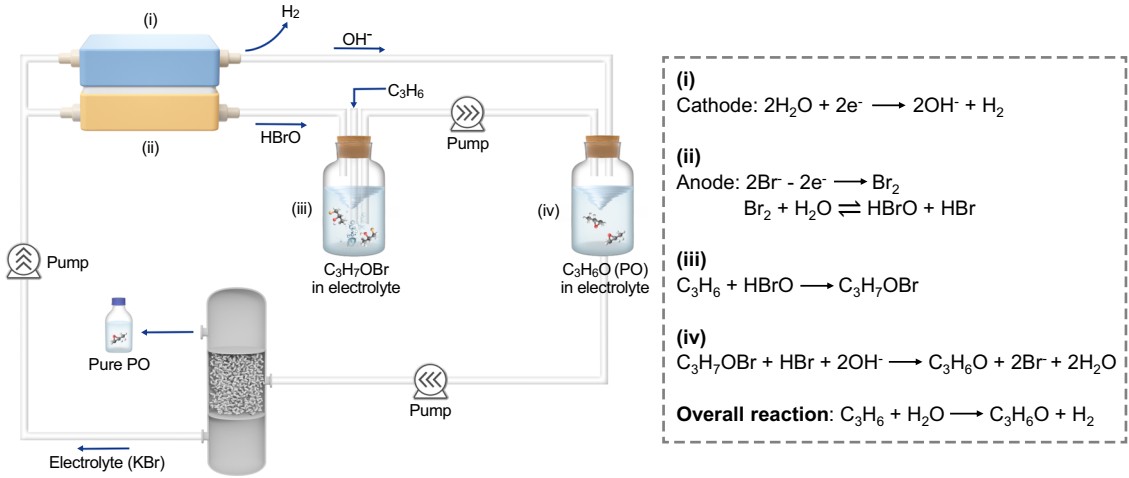

**(i)**
Cathode: $2H_2O + 2e^- \longrightarrow 2OH^- + H_2$

**(ii)**
Anode: $2Br^- - 2e^- \longrightarrow Br_2$
$Br_2 + H_2O \rightleftharpoons HBrO + HBr$

**(iii)**
$C_3H_6 + HBrO \longrightarrow C_3H_7OBr$

**(iv)**
$C_3H_7OBr + HBr + 2OH^- \longrightarrow C_3H_6O + 2Br^- + 2H_2O$

**Overall reaction**: $C_3H_6 + H_2O \longrightarrow C_3H_6O + H_2$

**Fig. 1 | Schematic illustration of the electrochemical bromohydrin route for efficient synthesis of PO.** Propylene was first converted to $C_3H_7OBr$ in an independent reactor through a bromide mediator; this product was then mixed with catholyte to form PO; pure PO was generated after the distillation operation and the residual electrolyte was able to be recycled without loss of Br$^-$.

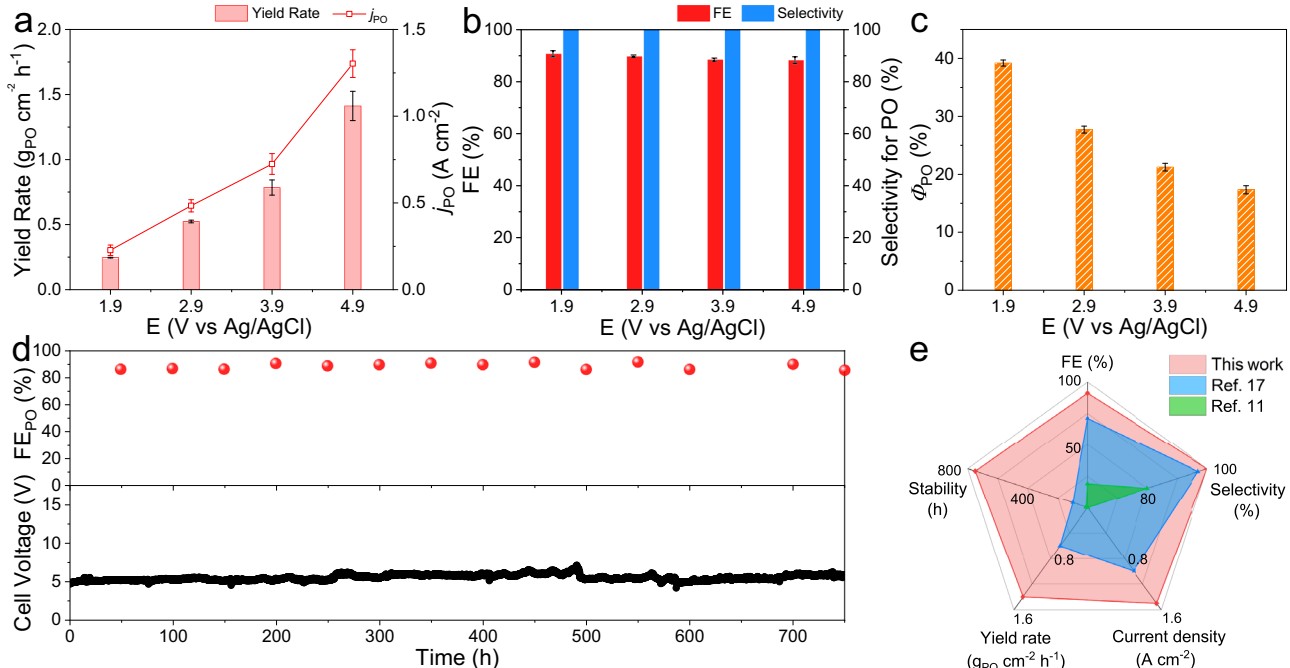

**Fig. 2 | Catalytic performance and long-term stability of the system. a–c** $j_{PO}$ and yield rate for PO (**a**), $FE_{PO}$ and selectivity for PO (**b**), and $\Phi_{PO}$ (**c**) at different applied potentials. **d** Stability test during 750-h (>30 days) continuous electrolysis at a constant current density of 250 mA cm$^{-2}$. **e** Comparison of FE, current density, selectivity, yield rate, and stability against currently reported electrochemical strategies for PO production. The error bars correspond to the standard deviation of three independent measurements.

bromohydrin route at the anode, we sought to explore the impact of Br$^-$ concentration on the catalytic performance. As shown in Supplementary Fig. 7, the $FE_{PO}$ was improved from 83% to 91% as the concentration of KBr was increased from 0.2 to 0.4 M. The enhanced $FE_{PO}$ could be ascribed to the facilitated mass diffusion of Br$^-$ at high concentrations. When the concentration of electrolyte was further increased to 5 M, the $FE_{PO}$ gradually decreased to 32%. This result validated that excessive Br$^-$ was detrimental to the reversible disproportionation reaction of Br$_2$ with water (Br$_2$ + H$_2$O ⇆ HBrO + HBr), thus inhibiting the formation of HBrO. Hence, the concentration of KBr was set to 0.4 M to evaluate the catalytic performance for the electrochemical transformation of propylene into PO.

When evaluating the performance, the geometrical current density of the carbon paper significantly enhanced with the increase of applied potentials (Supplementary Fig. 8). Figure 2a displays the partial current density for PO ($j_{PO}$) at different applied potentials. The $j_{PO}$ reached 1.3 A cm$^{-2}$ at 4.9 V vs Ag/AgCl, which far exceeded the industrially relevant current density (>200 mA cm$^{-2}$)[31,32]. We also calculated the yield rate of PO, which dramatically reached 1.4 g$_{PO}$ cm$^{-2}$ h$^{-1}$ at 4.9 V vs Ag/AgCl. Figure 2b shows the FE and the selectivity for PO. It is worth noting that the selectivity for PO reached above 99.9% at all applied potentials. No brominated products were detected (Supplementary Fig. 5a, b). This result could be attributed to the decoupled system, which avoided the direct contact between propylene and the high-concentration Br$_2$ at the anode surface. The $FE_{PO}$ was maintained above 88% at all applied potentials. In particular, when the applied potential was set to 1.9 V vs Ag/AgCl, the highest $FE_{PO}$ of 91% was achieved. In addition, the FE for C$_3$H$_7$OBr was close to $FE_{PO}$ at the corresponding applied potentials (Supplementary Fig. 9). To demonstrate whether high-valance bromine ions were produced, we tested the anion in the anolyte after electrolytic reaction via ion chromatography. As shown in Supplementary Fig. 10a, the BrO$_3^-$ or other high-valance bromine ions in the anolyte were below the detection limit. Besides, according to iodometric titration experiments, the FE for BrO$^-$ was determined to be 3.9% at 1.9 V vs Ag/AgCl (Supplementary

Fig. 10b). The FE for H$_2$ generated at the cathode was quantified via online GC, which corresponded to nearly 100% at all applied potentials (Supplementary Fig. 11). Given that the energy efficiency ($\Phi$) represents the conversion efficiency of electrical energy to target products, we further determined the $\Phi$ for PO ($\Phi_{PO}$) in this system[33,34]. As depicted in Fig. 2c, the maximum $\Phi_{PO}$ of 39% was attained at 1.9 V vs Ag/AgCl. We also measured the performance at different temperatures including 20, 30, and 40 °C to evaluate the temperature impact at high applied potentials. Interestingly, $FE_{PO}$ was maintained stable within the whole temperature range. At the same time, the $j_{PO}$ was gradually enhanced as the temperature increased (Supplementary Table 1). This phenomenon could be attributed to the lower electrolyzer impedance and the accelerated mass transfer of Br$^-$ at higher temperatures[35].

## Long-term stability of the system and the superiority of the electrochemical bromohydrin route

Apart from the activity and selectivity, the long-term stability of the system is another key parameter for industrial implementation[36,37]. The stability test was performed at a constant current density of 250 mA cm$^{-2}$. During a 750-h (>30 days) continuous measurement, the cell voltage exhibited negligible variation with relatively stable $FE_{PO}$ (only 0.73% decay), suggesting excellent long-term durability of the system (Fig. 2d). Moreover, the selectivity for PO was maintained above 99.9% during the continuous operation (Supplementary Fig. 12). Supplementary Fig. 13 illustrates that the morphology and structure of the carbon paper were perfectly preserved after the stability test. Notably, as shown by XPS results, apart from the peaks of C 1$s$ and O 1$s$, two new peaks ascribed to Br 3$d$ and 3$p$ were observed in the survey spectra of the carbon paper after the stability test (Supplementary Fig 14 and Supplementary Table 2). The Br species could mainly be assigned to the newly formed C-Br bonds on the carbon paper surface, which would prevent the carbon paper from being further oxidized[38–41]. It is worth mentioning that our system substantially outperformed the currently reported electrochemical strategies for PO production in FE, current density, selectivity, yield rate, and stability (Fig. 2e).

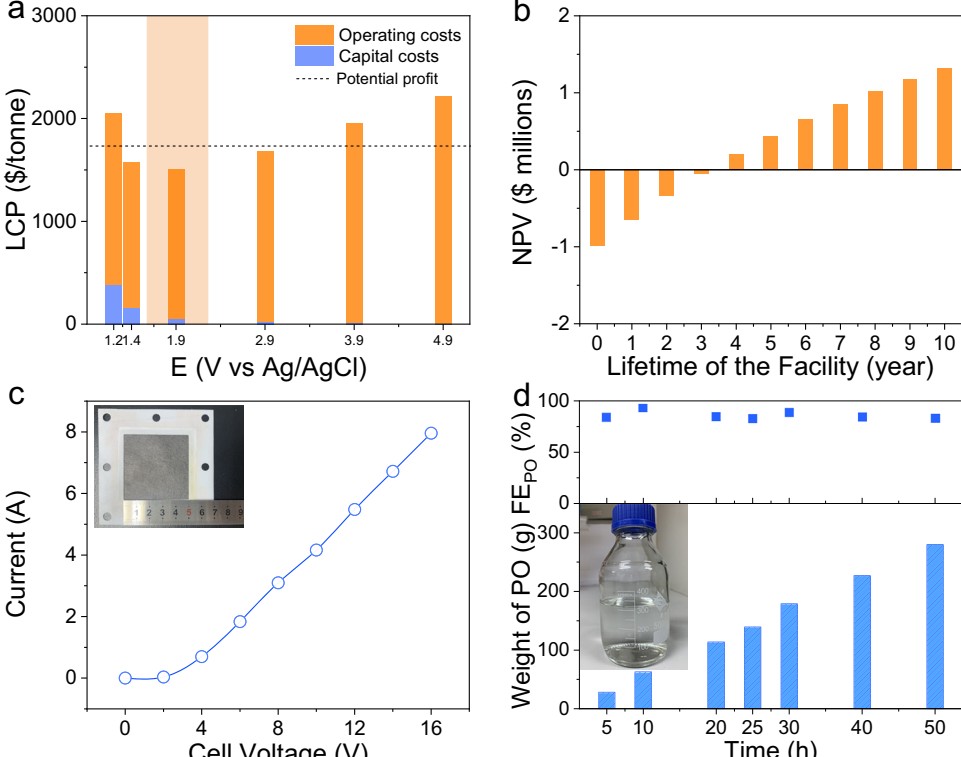

**Fig. 3 | TEA and large-scale production of PO. a** LCP of the electrochemical bromohydrin route at different applied potentials. **b** End-of-life NPV values of the electrochemical bromohydrin route at 250 mA cm⁻². **c** The steady-state current of the enlarged electrode with a geometric area of $5 \times 5$ cm$^2$ in 0.4 M KBr. The inset shows a photograph of the assembled electrode. **d** Accumulated PO in the electrolyte during the 50-h continuous galvanostatic electrolysis at an overall current of 6.25 A. The inset shows a photograph of the pure PO (315 mL) separated from the electrolyte.

To further clarify the superiority of the electrochemical bromohydrin route for the synthesis of PO, other electrochemical halohydrin routes using chloride and iodide as mediators were explored. Supplementary Fig. 15 depicts the LSV curves in 0.4 M KCl, KBr, and KI, respectively. The current density in KI was the highest due to the lowest standard electrode potential of I⁻ to $I_2$ (0.53 V vs RHE) among the three halogen ions. Nevertheless, no PO was detected once iodine was employed as the mediator for the electrochemical transformation of propylene. This result could be attributed to the sluggish disproportionation reaction of $I_2$ into HIO and the spontaneous decomposition of HIO ($3HIO \rightarrow 2HI + HIO_3$) in aqueous solution[42]. Supplementary Fig. 16 shows the FE$_{PO}$ at various applied potentials via the electrochemical chlorohydrin route. At 1.9 V vs Ag/AgCl, the FE$_{PO}$ was only 56%, proving the superiority of the electrochemical bromohydrin route relative to electrochemical chlorohydrin route. To demonstrate the competition of OER, the generated oxygen ($O_2$) was detected and quantified by an online GC equipped with a thermal conductivity detector (TCD). Obviously, the signal intensity of $O_2$ generated via the electrochemical bromohydrin route was smaller than that via the electrochemical chlorohydrin route at all applied potentials (Supplementary Fig. 17a, b). The highest FE for $O_2$ via the electrochemical bromohydrin route was only 2.2% whereas the lowest FE for $O_2$ via the electrochemical chlorohydrin route reached 16.4% (Supplementary Fig. 17c, d). Additionally, the volatility of $Br_2$ and $Cl_2$ in electrolytes was also one of the key parameters affecting the catalytic performance. We set two absorption devices containing 3 M NaOH to capture the volatile $Br_2$ and $Cl_2$ in the gas products. To determine the existence of Br⁻ and Cl⁻, the above solutions were adjusted to neutral with $HNO_3$ before the addition of $AgNO_3$. As displayed in Supplementary Fig. 18, the absorption solution of $Br_2$ remained clear whereas white precipitation (AgCl) was observed in the absorption solution of

$Cl_2$, suggesting the negligible volatilization of $Br_2$ and the severe loss of $Cl_2$.

## The practical application possibility of the electrochemical bromohydrin route

To investigate the industrial feasibility of the electrochemical bromohydrin route, we conducted a techno-economic analysis (TEA) to determine the levelized cost of the product (LCP) and the end-of-life net present value (NPV)[17,43–46]. Based on the current market price of propylene, PO, and $H_2$, we estimated capital costs and operating costs at potentials ranging from 1.2 to 4.9 V vs Ag/AgCl (Supplementary Fig. 19). Figure 3a shows the LCP at different applied potentials. When the applied potential was set at 1.9 V vs Ag/AgCl with an average current density of about 250 mA cm⁻², the LCP was the most economical. Sensitivity analysis revealed that electrochemical parameters had a significant impact on the LCP, such as applied potential, current density, and FE$_{PO}$, which should be considered carefully to reduce the overall costs (Supplementary Table 3 and Supplementary Fig. 20). Considering that the end-of-life NPV is positively correlated with the LCP, we calculated the end-of-life NPV at the optimal current density of 250 mA cm⁻². As shown in Fig. 3b, the end-of-life NPV became profitable in the fourth year. These results demonstrated the economic feasibility of the electrochemical bromohydrin route.

On the basis of the above analyses, the practical application possibility of the electrochemical bromohydrin route was further assessed by amplifying the size of the electrode. Here, an enlarged two-electrode flow reactor with a geometric electrode area of 25 cm$^2$ was developed to scale up the production of PO (Supplementary Fig. 21 and inset of Supplementary Fig. 3c). Figure 3c shows the steady-state current at different applied potentials. When the current density was set as 250 mA cm⁻² (6.25 A), the single-pass conversion of propylene

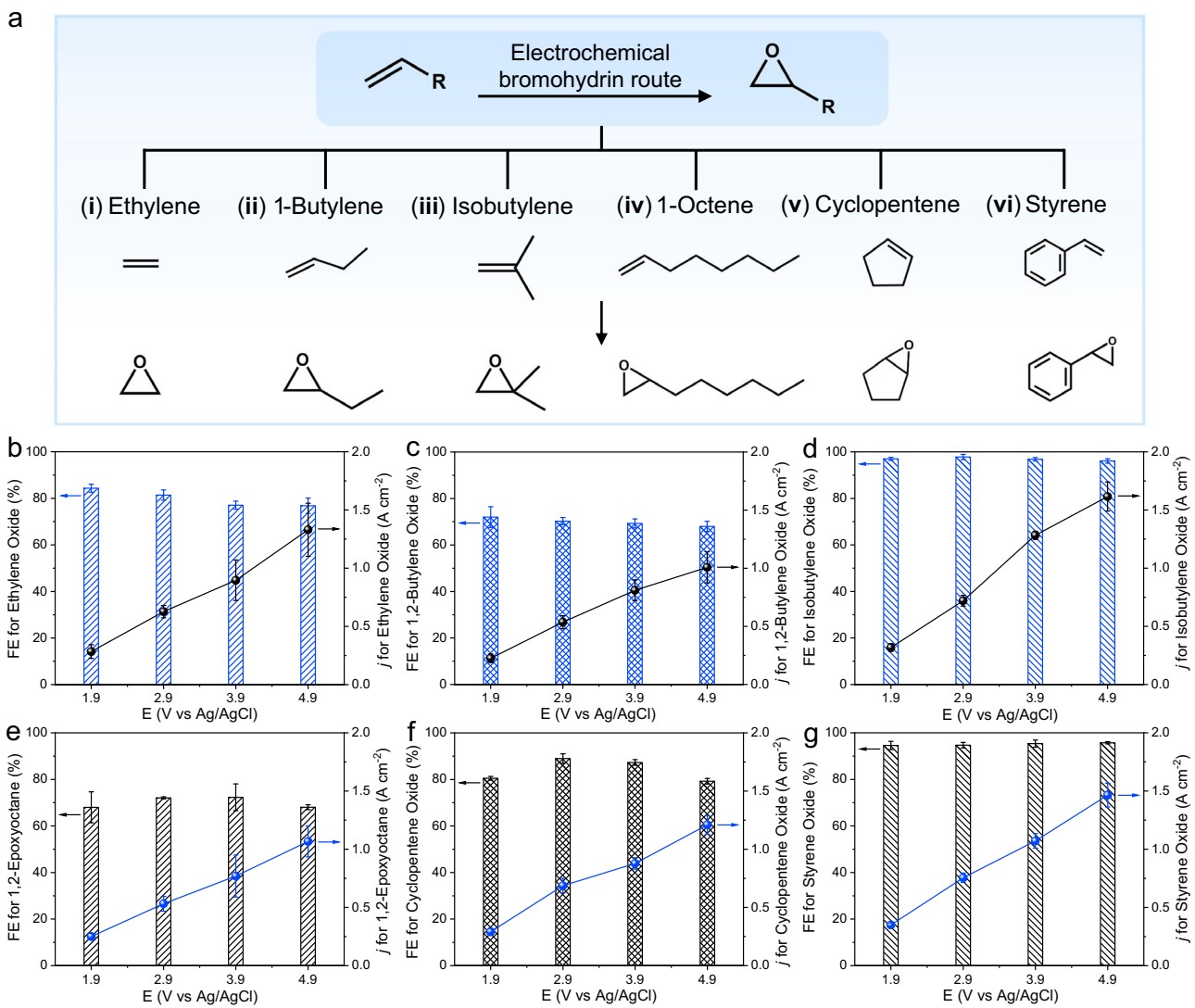

**Fig. 4 | The universality of the electrochemical bromohydrin route for other alkene substrates. a** Schematic diagram of the universality of the electrochemical bromohydrin route for a wide range of alkene substrates. **b–d** FE and partial current density for ethylene oxide (**b**), 1,2-butylene oxide (**c**), and isobutylene oxide (**d**) at different applied potentials. **e–g** FE and partial current density for 1,2-epoxyoctane (**e**), cyclopentene oxide (**f**), and styrene oxide (**g**) at different applied potentials. The error bars correspond to the standard deviation of three independent measurements.

reached up to 66% at a gas flow rate of 60 SCCM (Supplementary Fig. 22). We conducted 50-h continuous galvanostatic electrolysis at 6.25 A for the preparation of PO. With the increase of reaction time, the yield for PO was gradually enhanced with $FE_{PO}$ maintaining higher than 83% (Fig. 3d). Finally, 280 g of PO was accumulated in the electrolyte by determining the concentration of PO. To separate pure PO from the electrolyte for commercial application, we designed a vacuum distillation unit to prevent the hydrolysis of PO at high temperatures (Supplementary Fig. 23). As depicted in Fig. 3d, 262 g (315 mL) of pure PO was obtained with a considerable distillation yield of 93.6%. As shown by the $^1H$ NMR and $^{13}C$ NMR spectra in Supplementary Fig. 24, the generated PO possessed comparable purity with respect to commercial PO. As an approximation of the industrial manufacturing process, the electrochemical bromohydrin route is highly desirable to further push forward the commercialized production of PO.

### The universality of electrochemical bromohydrin route for other alkenes
To investigate the universality of the electrochemical bromohydrin route, we attempted to apply this approach to a wide range of alkene substrates. As illustrated in Fig. 4a, a series of gaseous and liquid alkene

substrates were explored. For gaseous alkenes including ethylene, 1-butylene, and isobutylene, at 1.9 V vs Ag/AgCl, the FEs for ethylene oxide, 1,2-butylene oxide, and isobutylene oxide were 84%, 70%, and 98%, respectively (Fig. 4b–d). In addition to gaseous alkenes, the electrochemical bromohydrin route was also applicable for the transformation of liquid alkenes. As shown in Fig. 4e–g, linear, cyclic, and aromatic liquid alkenes were also examined including 1-octene, cyclopentene, and styrene. At 1.9 V vs Ag/AgCl, the FEs for 1,2-epoxyoctane, cyclopentene oxide, and styrene oxide were 70%, 81%, and 93%, respectively. These FEs for corresponding epoxides slightly fluctuated when the applied potentials ranged from 1.9 to 4.9 V vs Ag/AgCl. Furthermore, the partial current density for the corresponding epoxides mentioned above all satisfied the standard of industrially relevant current density. These results undoubtedly validated the feasibility of the electrochemical bromohydrin route for the synthesis of epoxides.

### Discussion
In summary, we developed a spatial decoupling system using bromide as the mediator for the efficient electrosynthesis of PO. This route achieved record-high current density, FE, and product selectivity

relative to other electrochemical processes. Moreover, this system is expected to be scaled up and could be widely applied to diverse alkene substrates. Our work provides an alternative route for PO production, which overcomes the challenges in the chlorohydrin, hydrogen peroxide-based, and other electrochemical processes, thereby paving the way towards the electrification of chemical manufacturing.

## Methods

### Chemicals and materials

Carbon papers (TGP-H-090) were purchased from Toray Industries, Inc (Tokyo, Japan). Potassium bromide (KBr, 99%), potassium chloride (KCl, 99.5%), potassium iodide (KI, 99%), sodium hydroxide (NaOH, 96%), nitric acid (HNO$_3$, 65-68%), silver nitrate (AgNO$_3$, 99.8%) and hydrochloric acid (HCl, 36-38%) were all purchased from Sinopharm Chemical Reagent Co. Ltd. (Shanghai, China). PO (99.7%), propylene bromohydrin (75%), ethylene oxide (99.5%), 1,2-butylene oxide (99%), isobutylene oxide (97%), 1,2-epoxyoctane (97%), cyclopentene oxide (97%), and styrene oxide (97%) were purchased from Aladdin Co. Ltd. (Shanghai, China). 4,4-dimethyl-4-silapentane−1-sulfonic acid (DSS), dimethyl sulfoxide-$d_6$ (DMSO-$d_6$, 99.9 atom% D), and Nafion 115 membrane were purchased from Sigma-Aldrich. The deionized (DI) water with a resistivity of 18.2 MΩ cm was provided by a Millipore Milli-Q grade. All of the chemicals were used without any further purification.

### Electrochemical measurements

For the electrochemical transformation of propylene into PO, the electrochemical measurements were carried out in a three-electrode H-cell equipped with inlets and outlets for the electrolyte in the anodic and cathodic chambers, respectively. The anodic and cathodic chambers with volumes of 36 mL were separated by Nafion 115 membrane. Titanium (Ti) foil with a geometric area of 9 cm$^2$ and Ag/AgCl (3 M KCl) served as the counter electrode and reference electrode, respectively. Ti foil was first etched in boiled 6 M HCl for 30 min before conducting the electrochemical measurements. Carbon paper with a geometric area of 0.5 cm$^2$ was used as the working electrode. The potentials were controlled by an Autolab potentiostat/galvanostat (CHI 1140E). All potentials were measured against the Ag/AgCl reference electrode. The measurements were performed without iR compensation. Tafel slope was determined by fitting polarization curves data to the Tafel equation:

$$E = a + b \log |j| \tag{1}$$

$E$ is the applied potential, $b$ is the tafel slope, and $j$ is the current density.

$b$, the tafel slope.

$j$, the current density.

The FE and selectivity measurements were operated at 1.9, 2.9, 3.9, and 4.9 V vs Ag/AgCl in an environmental chamber. The chronoamperometric electrolysis was performed at each potential for 30 min. The electrolyte was circulated through the cell using peristaltic pumps. The anolyte after electrolysis was pumped out of the cell to the separated reactor using peristaltic pumps. The propylene gas kept bubbled into the separated reactor at a flow rate of 20 SCCM during the electrolysis process. To increase the contact interface between propylene and anolyte, we dispersed the propylene into dense bubbles by utilizing a sand core airway accompanied by vigorous stir. At the end of electrolysis, the anolyte and catholyte were mixed in equal proportion to generate PO through a saponification process. Subsequently, the products in the anolyte and mixed electrolyte were quantified via 400 MHz $^1$H NMR spectrometer. Typically, 0.4 mL of the electrolyte after electrolysis was mixed with 0.1 mL of DMSO-$d_6$ and 0.1 mL of 6 mM DSS solution. The gaseous products were monitored via an online GC equipped with a TCD.

For long-term stability test for the electrochemical synthesis of PO, the electrochemical measurement was conducted at 250 mA cm$^{-2}$ in 0.4 M KBr electrolyte at the environmental temperature (25 °C). The electrolyte was circulated through the cell at a flow rate of 20 mL min$^{-1}$ using peristaltic pumps. The propylene gas kept bubbling into the separated reactor at a flow rate of 20 SCCM during the electrolysis process. During the electrolysis process, the mixed electrolyte was taken out at regular intervals for quantitative analysis via $^1$H NMR.

The large-scale production of PO was performed with a geometric electrode area of 25 cm$^2$. Ti foil with a geometric area of 25 cm$^2$ served as the counter electrode. The electrochemical measurement was conducted at an overall current of 6.25 A in 0.4 M KBr electrolyte. The current was controlled by a Booster 2050 (Corrtest, China). The anolyte and catholyte were pumped out of the cell using peristaltic pumps. The propylene gas kept bubbling into the separated reactor at a flow rate of 60 SCCM during the electrolysis process. The mixed electrolyte was taken out at regular intervals for quantitative analysis via $^1$H NMR.

### Product separation

As the selectivity for PO was above 99.9%, PO could be separated from the electrolyte through binary distillation. To separate the pure PO from the electrolyte for commercial application, we designed a vacuum distillation unit to prevent the hydrolysis of PO at high temperatures. The length of the lab-scale packed distillation column was 1000 mm, which was filled with glass springs with a height of 800 mm. In a typical experiment, the mixed electrolyte was introduced into the middle of the packed distillation column at a flow rate of 50 mL min$^{-1}$, which then flowed into the reboiler and heated to generate sufficient steam to exchange heat with fresh pumped feed. To improve the thermal efficiency, the excess reboiling liquid was pumped out to undergo heat exchange with the cold feed. The distillation pressure was maintained at 50 kPa. Under this pressure, the boiling point of PO and water was 16 and 82 °C, respectively. In this case, the liquid in the reboiler was maintained at a temperature slightly higher than 82 °C to generate steam continuously. As for the condensation process, a pre-reflux unit with a cooling water of 25 °C was installed on the top of the distillation column to reduce the load of the condenser and enhance the reflux efficiency. The condenser was set at −15 °C to collect the vapor of PO.

### Performance calculations

The FE for liquid products was calculated by a given equation as follows:

$$FE_{liquid}(\%) = C \times V \times N \times F / Q \tag{2}$$

$C$ is the concentration of liquid products, $V$ is the volume of the electrolyte, $N$ is the number of electrons transferred for product formation, $F$ is the Faraday constant, 96485 C mol$^{-1}$, $Q$ is the quantity of electric charge integrated by the potentiostat.

The FE for gaseous products was computed using the following formula:

$$FE_{gas}(\%) = N \times F \times x \times S / j_{total} \tag{3}$$

$N$ is the number of electrons transferred for product formation, $F$ Faraday constant, 96485 C mol$^{-1}$, $x$ is the mole fraction of gaseous products, $S$ is the total molar flow rate of gas, $j_{total}$ is the total current.

The selectivity for PO was calculated as follows:

$$PO\ selectivity(\%) = n_{PO} / n_{products} \tag{4}$$

$n_{PO}$ is the amount of generated PO (mol) and $n_{products}$ is the total amount of liquid products (mol),

The $\Phi_{PO}$ was calculated according to the equation:

$$\Phi_{PO}(\%) = (FE \times \Delta E^0_{PO}) / \Delta E_{PO} \qquad (5)$$

FE is the Faradaic efficiency for PO, $\Delta E^0_{PO}$ is the difference between the standard half reaction potentials for the oxidation of $Br^-$ into $Br_2$ (1.08 V vs RHE) and hydrogen evolution reaction (HER, 0 V vs RHE), $\Delta E_{PO}$ is the difference between the working potential at the anode and the standard potential for HER.

## Instrumentations

XRD patterns were recorded by using a Philips X'Pert Pro Super diffractometer with Cu-$K\alpha$ radiation ($\lambda = 1.54178$ Å). Scanning electron microscopy (SEM) images were taken using a Hitachi SU8220 scanning electron microscope. High resolution transmission electron microscope (HRTEM) was carried out on a field-emission transmission electron microscope (JEOL ARM-200F) operating at 200 kV accelerating voltage. SAED was carried out on a JEOL ARM-200F field-emission transmission electron microscope operating at an accelerating voltage of 200 kV using Cu-based TEM grids. The Raman spectrum was conducted via LabRAM HR Evolution (Horiba) Roman system with a 532 nm excitation laser. The liquid products were examined on a Varian 400 MHz NMR spectrometer (Bruker AVANCE AV III 400). The gaseous products were detected via online gas chromatography (GC2014, Shimadzu, Japan).

## Data availability

The source data underlying Figs. 1–4 and Supplementary Figs. 1–24 generated in this study are provided as a Source Data file. Source data are provided with this paper.

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

## Acknowledgements
Z. Geng acknowledges Strategic Priority Research Program of the Chinese Academy of Sciences (XDB0450401), National key R&D Program of China (2022YFC2106000), NSFC (22322901), CAS project for young scientists in basic research (YSBR-022), and Fundamental Research Funds for the Central Universities (WK9990000114). J. Zeng acknowledges National Key Research and Development Program of China (2021YFA1500500 and 2019YFA0405600), National Science Fund for Distinguished Young Scholars (21925204), NSFC (22221003 and 22250007), CAS project for young scientists in basic research (YSBR-051), Collaborative Innovation Program of Hefei Science Center, CAS (2022HSC-CIP004), the Joint Fund of the Yulin University and the Dalian National Laboratory for Clean Energy (YLU-DNL Fund 2022012), International Partnership Program of Chinese Academy of Sciences (123GJHZ2022101GC), Fundamental Research Funds for the Central Universities, and USTC Research Funds of the Double First-Class Initiative (YD9990002014). Y. Liu acknowledges NSFC (22209161). J. Ke acknowledges the China Postdoctoral Science Foundation (2023TQ0338 and 2023M743368), and Postdoctoral Fellowship Program of CPSF. This work was partially carried out at the Instruments Center for Physical Science, University of Science and Technology of China.

## Author contributions
Z. Geng and J. Zeng supervised this project. M. Chi, J. Ke, and Y. Liu performed most of the experiments and analyzed the experimental data. M. Wei provided help in the product separation. H. Li, J. Zhao, and Y. Zhou conducted characterizations and analyzed the results. M. Chi, Z. Gu, Z. Geng, and J. Zeng wrote the manuscript. All authors discussed the results and assisted during manuscript preparation.

## Competing interests
The authors declare no competing interests.
