## [Peer review file · Nature Communications]

REVIEWER COMMENTS

Reviewer #1 (Remarks to the Author):

In this paper, decoupling strategy is applied to prevent overoxidation of propylene and initiate in-direct conversion of propylene-to-propylene oxide (PO). Accordingly, the authors achieve propylene-to-PO with 100% of selectivity and long stability at 250 mA cm⁻². However, this generally in-direct method has been used in many works for alkene oxidation (ACS Catal. 2020, 10, 14015–14023; ACS Energy Lett. 2019, 4, 600–605; Angew. Chem. Int. Ed. 2023, e202311570, etc.). Of those works, mechanism and factors of this strategy have investigated in detail. Therefore, this work is short of insight and novelty. Some major technical comments are shown here.

1. Since the bromohydrin route involves only a single electrochemical step, i.e. Br⁻ oxidation, the calculation of PO partial current density and FE is questionable. This calculation primarily reflects the electrochemical Br⁻ oxidation performance rather than accurately representing the PO production performance.
2. In Figure 1, mechanism indicates that anode chamber will generate HBrO spontaneously from Br₂. HBrO will further hydrolyse into other weak acids and occur irreversible cleavage causing FE loss and pH deviation rapidly in neutral electrolyte. As a result, potentials (vs RHE) in this work remain stable is impractical.
3. In Figure 2b, the authors should provide proof to evidence the missing FE. The byproducts of the system need to be detected to confirm the feasibility of the bromohydrin route, for example brominated organics.
4. In Figure 2, this work requires high potentials (voltages) to trigger reaction. Numerous reports show low potential (only 2~3 V) demanding for alkene oxidation adopting this strategy. From energy-saving and cost perspective, energy efficiency (energy utilization) should be considered.
5. OER is a strong competition reaction to bromine evolution reaction in such low KBr electrolyte (0.4 M). The thermodynamic oxidation potentials of Br⁻ and Cl⁻ are far lower than the actual applied potentials (2.5~5.5 V vs RHE), so that OER competition can significantly occur on the electrode regardless of the electrolyte anion. Then it is surprising that O₂ was almost not detected in the bromohydrin route as compared to the chlorohydrin route. The reason needs to be further explained. The authors should provide detailed evidence to prove almost entire OER inhibition in this electrolyte under very high potential. In addition, KBr concentration is generally sensitive to bromine evolution reaction. Why do authors select 0.4 M KBr as electrolyte.
6. Is there possibility that the improved FE compared with the chlorohydrin route is related to the inhibited HBrO dissociation? Is there any solubility issue associated with propylene when it reacts with HBrO?

7. It is suggested to verify the conversion and yield rates of the chemical steps involved in the formation of C₃H₇BrO and PO. This verification could be achieved, for example, by introducing HBrO solution to confirm the production of PO.

8. The FE of generated H₂ at the cathode and the catholyte/anolyte pH levels were not mentioned in the manuscript. After the Br₂ disproportionation reaction, it is expected that the anolyte could be strongly acidic so may need substantial OH⁻ from the cathode to neutralize it. Thus, it is important to determine the OH⁻ produced in the cathode where the catholyte pH could be an important factor to limit the kinetics of PO formation step.

9. In TEA model, reasonable references should be added. The electrolyser costs and prices of feedstock and products should re-calculate and re-set according to current market reports, respectively. Why parameters of model (full cell voltage, current density, etc.) are not based on experimental results? TEA results should be re-calculated.

Reviewer #2 (Remarks to the Author):

The manuscript by Zeng et al. reported an electrochemical bromohydrin route for highly efficient synthesis of propylene oxide. The strategy integrated electrolysis and propylene by coupling electrolysis of bromide and propylene conversion process within separated reactors. The system demonstrates the capability for continuous propylene oxide production over a month at high current density, maintaining both high selectivity and Faradaic efficiency. While the work is intriguing and showcases the potential for large-scale epoxide production to replace traditional methods, the manuscript could benefit from the inclusion if there are more fundamental and mechanistic data. Strengthening the manuscript with additional principal data would better align with the requirements of a scientific paper, rather than solely focusing on the technological process and its results, if the manuscript will be published in this journal.

Nevertheless, the manuscript could potentially benefit from the following suggestions.

1. Whether the authors have investigated other commercial electrode materials besides carbon paper. The detail reaction mechanism involved with Br⁻ on carbon paper would help to improve understanding and thoughtful of the manuscript.

2. It is essential to provide the utilization rate of substrate in this reaction system, particularly for gas substrate, considering the high cost.

3. To reflect the obvious advantages of the electrochemical bromohydrin route, the author could make a cost-comparison with traditional industrial process, such as H₂O₂ and O₂.

4. The assertion of “breaking the limit of the current” is overstatement due to the decreased energy efficiency at elevated working potentials. What does the high temperature of reactors resulting from high cell voltage impact the reaction?

Reviewer #3 (Remarks to the Author):

In this work, the authors developed an impressive strategy by spatially decoupling the electrolysis process and the propylene conversion process for efficient propylene oxide (PO) production. This strategy achieved record-high FE (91%), product selectivity (100%), and long-term stability (operation for >30 days) relative to other electrochemical processes. In addition, the authors designed an enlarged flow reactor with a geometric electrode area of 25 cm² to scale up the manufacture of PO. Combined with the designed separation device, the practical application promise of this strategy was demonstrated. Finally, this strategy could also be applied to the efficient transformation of other alkenes, including gaseous and liquid alkenes. This work is impressive and well-organized in general. Therefore, I recommend the publication of this manuscript on Nature Communications after addressing a few minor questions.

-In this strategy, the concentration of bromine ions may affect the reaction efficiency. It is necessary to provide the experimental results for optimizing the concentration of bromine ions.

-Bromine may also be further oxidized to other high valence products such as bromic acid or added to organic substrates to produce brominated products. The formation of these by-products should be excluded.

-The FE for H₂ in the whole reaction process was not measured, so it was not rigorous to directly assume the 100%-FE for H₂ when calculating techno-economic analysis (TEA). Therefore, the authors are advised to determine the FE for H₂ in the whole reaction process to calculate the TEA.

-In the TEA section, the assumption on the price of electricity “which is higher than the price of industrial electricity” was presented without sufficient supporting reference. More references should be listed to support the assumption.

-The post-reaction structure of the carbon paper was not well characterized. More characterizations of the spent catalyst after the durability test should be offered.

-The experimental evidence about the reaction mechanism was limited. I suggest the authors supplement relevant experiments to verify the reaction mechanism.

-The discussion about Faradaic efficiency was presented without much context or explanation. In-depth discussion and explanation of this result is necessary.

-Some figure captions are unclear, such as the Supplementary Figure 2 and Supplementary Figure 5. The authors should check and improve the content of the figure captions carefully.

-Some experimental details should be supplemented. For example, for the H-cell experiments, the volume of the cell was not mentioned. A thorough check and proofread is necessary.

Point-by-point response to reviewers' comments

Manuscript ID: NCOMMS-23-51936

MS Type: Article

Title: "Linking propylene and anode via bromide mediator in separated reactors for high-current and stable synthesis of propylene oxide"

Reviewer #1:

"In this paper, decoupling strategy is applied to prevent overoxidation of propylene and initiate in-direct conversion of propylene-to-propylene oxide (PO). Accordingly, the authors achieve propylene-to-PO with 100% of selectivity and long stability at 250 mA cm⁻². However, this generally in-direct method has been used in many works for alkene oxidation (ACS Catal. 2020, 10, 14015-14023; ACS Energy Lett. 2019, 4, 600-605; Angew. Chem. Int. Ed. 2023, e202311570, etc.). Of those works, mechanism and factors of this strategy have investigated in detail. Therefore, this work is short of insight and novelty. Some major technical comments are shown here."

We sincerely thank this reviewer for his/her careful reading of our manuscript. We acknowledge that the in-direct method has indeed been reported in other works. However, their performance (including activity, stability, and selectivity) still suffers from the following challenges. Firstly, the operation current was limited (<0.35 A). Specifically, the alkenes were bubbled or placed into the electrolyzer directly in previous works. In this case, the alkenes inevitably underwent unwanted side reactions at high currents such as overoxidation, which was not conducive to PO production at high currents, thus restricting the scalable production of PO. At the same time, the side reactions caused the loss of alkenes, resulting in the rising cost of industrial manufacture. Secondly, the current stability in previous works was impeded by the massive input of alkene and vigorously stirred electrolytes. Thirdly, the FE was limited since the generated HBrO was proved to suffer from cleavage reactions when exposed to metal-based anodes with specific adsorption sites. Fourthly, 1,2-dichloropropane (C₃H₆Cl₂) was regarded as a main side product of the traditional chlorohydrin method, which was derived from the addition reaction between Cl₂ and propylene. In the same way, halogenated products might be produced in previous works since the local concentration of Cl₂ or Br₂ at the anode surface was relatively high. The generation of brominated products restricted the reaction selectivity, leading to higher product separation costs.

In our work, to avoid the above restrictions, we spatially decoupled the electrolysis process and the propylene conversion process by utilizing the bromide mediator to link propylene and anode within separated reactors. Notably, the spatially decoupled system avoided direct contact between the electrode and propylene, thereby eliminating both the influence of high current on propylene and the interference of system stability caused by massive feedstock input into the electrolyzer. In addition, the cleavage of HBrO could be weakened since the electrolyte would not be exposed to electrodes for a long time owing to the decoupled system. The inert carbon anode without adsorption sites could also alleviate the cleavage. Moreover, propylene would not contact with high-concentration Br₂ at the anode surface, thereby inhibiting the generation of brominated products (1,2-dibromopropane, C₃H₆Br₂). Based on the above efforts, we realized the high-current and scalable synthesis of PO at 6.25 A with an enlarged working electrode area of

25 cm², far larger than previous works (<1.1 cm²). The FE_{PO} achieved 91% along with the >99.9% selectivity and excellent stability (750 h). Combined with a designed separation device, pure PO was obtained. Besides, our work investigated the applicability of multiple substrates including both gaseous alkenes (such as ethylene, 1-butylene, and isobutylene) and liquid alkenes (such as 1-octene, cyclopentene, and styrene).

All in all, we believe that the combination of superior performance, scalable production, and downstream separation has a certain impetus for the electrification production of PO, which fits well with the scope of *Nature Communications*. Prompted by the insightful suggestions from this reviewer, we have revised our abstract, and introduction sections to further emphasize our theme of developing a feasible route for the high-current and scalable production of PO by spatially decoupling the electrolysis process and the propylene conversion process via the bromide mediator. We have also added the literature mentioned by this reviewer in the revised manuscript.

“1. Since the bromohydrin route involves only a single electrochemical step, i.e. Br⁻ oxidation, the calculation of PO partial current density and FE is questionable. This calculation primarily reflects the electrochemical Br⁻ oxidation performance rather than accurately representing the PO production performance.”

We genuinely thank this reviewer for his/her valuable comments. The FE for Br⁻ electrooxidation could be calculated according to the concentration of generated Br₂ as follows:

$$\text{FE (\%)} = C \times V \times N \times F / Q \quad (1)$$

C, the concentration of generated Br₂.

V, the volume of the electrolyte.

N, the number of electrons transferred for product formation, which is 2 for Br₂.

F, the Faraday constant.

Q, the quantity of electric charge integrated by the *i-t* curve.

Actually, the generated Br₂ was easy to disproportionate into HBrO, which reacted with the propylene to bromohydrin immediately. Since Br₂ was unstable in the aqueous solution, we didn't calculate the FE for Br₂ in the manuscript.

Notably, the Br₂ returned to the original state of Br⁻ at the end of the reaction, suggesting that the Br⁻ only served as a mediator for propylene conversion. The reaction equation for PO production could be described as follows:

Consequently, as demonstrated by the calculation equation in our manuscript, we calculated the FE and partial current density for PO based on the concentration of generated PO. For the indirect reaction system, this performance evaluation method was reasonable, which has also been widely employed in many works [*Science* **368**, 1228-1233 (2020); *Angew. Chem. Int. Ed.* **62**, e202311570 (2023)]. Moreover, the Br₂ generated from the Br⁻ electrooxidation could transform into multiple products, such as C₃H₇OBr, 1,2-dibromopropane (C₃H₆Br₂), and the unreactive BrO[•]. In this case, the FE for Br₂ was higher than FE_{PO}. Therefore, the calculation of PO partial current density and FE based on the concentration of generated PO could accurately reflect the performance of PO rather than Br₂.

“2. In Figure 1, mechanism indicates that anode chamber will generate HBrO spontaneously from Br₂. HBrO will further hydrolyse into other weak acids and occur irreversible cleavage causing FE loss and pH deviation rapidly in neutral electrolyte. As a result, potentials (vs RHE) in this work remain stable is impractical.”

We sincerely thank this reviewer for pointing out this issue. The potentials in the previous manuscript were measured against the Ag/AgCl reference electrode and converted to the RHE reference scale on account of the equation:

$$E \text{ (vs RHE)} = E \text{ (vs Ag/AgCl)} + 0.21 \text{ V} + 0.0591 \times \text{pH} \quad (3)$$

However, the disproportionation of Br₂ generated a molecule of HBrO and a molecule of HBr. As a strong acid, HBr could dissociate completely into H⁺ and Br⁻, which would cause pH deviation of the electrolyte. The pH value of the anolyte dropped from 6.80 to 2.16 after electrolysis, as determined by a pH meter. Considering the pH deviation of the electrolyte, it is indeed inaccurate to directly convert the Ag/AgCl reference scale to the RHE reference scale. In the revised manuscript, we have revised all of the potentials with Ag/AgCl as the reference electrode.

“3. In Figure 2b, the authors should provide proof to evidence the missing FE. The byproducts of the system need to be detected to confirm the feasibility of the bromohydrin route, for example brominated organics.”

We genuinely thank this reviewer’s constructive suggestion. The missing FE could be ascribed to the following reasons: (1) the competitive OER, (2) the overoxidation of Br⁻ to other high-valance bromine ions, (3) the volatilization of generated Br₂, (4) the generation of other brominated organics, and (5) the irreversible cleavage of HBrO. In the previous manuscript, we determined the FE for O₂, which was 0.8% at 1.9 V vs Ag/AgCl (Supplementary Fig. 17). To determine whether high-valance bromine ions were produced, we tested the anion in the anolyte after electrolytic reaction via ion chromatography in the revised manuscript. As shown in Supplementary Figure 9a, the BrO₃⁻ or other high-valance bromine ions in the anolyte were below the detection limit. As for scenario (3), the capture experiment in our previous manuscript indicated that the Br₂ volatilization was almost negligible. We also explored the possibility of Br₂ engaging in an addition reaction with propylene to produce brominated products (1,2-dibromopropane, C₃H₆Br₂) in the previous manuscript. As determined by ¹H NMR, the byproducts after the addition reaction were below the detection limit (Supplementary Fig. 5a-b). This result could be attributed to the decoupled system, which avoided the direct contact between propylene and the high-concentration Br₂ at the anode surface. To investigate scenario (5), we have conducted iodometric titration experiments to determine the amount of BrO⁻ in the anolyte after the reaction. As shown in Supplementary Figure 9b, the FE for BrO⁻ was 3.9% at 1.9 V vs Ag/AgCl. Therefore, the missing FE could be mainly ascribed to the OER and the cleavage of HBrO. We have added the corresponding discussions in the revised manuscript (lines 6 to 9 and lines 12 to 16 on page 6, highlighted in yellow color).

“4. In Figure 2, this work requires high potentials (voltages) to trigger reaction. Numerous reports show low potential (only 2~3 V) demanding for alkene oxidation adopting this strategy.

From energy-saving and cost perspective, energy efficiency (energy utilization) should be considered.”

We sincerely thank this reviewer for raising this issue. When evaluating the overpotential of an electrochemical system, the current density should be specified [*Nat. Sustain.* **6**, 236-238 (2023)]. The partial current densities in many reports were lower than 100 mA cm⁻², thus showing low potentials of only 2~3 V. However, In our work, we achieved a j_{PO} of 227 mA cm⁻² at 1.9 V vs Ag/AgCl (previous 2.5 V vs RHE). To further evaluate the feasibility of our system at higher current densities, we continued to increase the applied potential to 2.9, 3.9, and 4.9 V vs Ag/AgCl. At 4.9 V vs Ag/AgCl, the FE_{PO} was still maintained at 88% with a j_{PO} of 1303 mA cm⁻².

At the same time, in the TEA, we have considered energy efficiency to evaluate the optimal operating condition since we conducted the calculations based on our experimental results including the j and cell voltage. According to the TEA result, the cost was optimal at 250 mA cm⁻², with an energy efficiency of 39%. Consequently, we then conducted the stability test and scalable production at 250 mA cm⁻².

“5. OER is a strong competition reaction to bromine evolution reaction in such low KBr electrolyte (0.4 M). The thermodynamic oxidation potentials of Br⁻ and Cl⁻ are far lower than the actual applied potentials (2.5~5.5 V vs RHE), so that OER competition can significantly occur on the electrode regardless of the electrolyte anion. Then it is surprising that O₂ was almost not detected in the bromohydrin route as compared to the chlorohydrin route. The reason needs to be further explained. The authors should provide detailed evidence to prove almost entire OER inhibition in this electrolyte under very high potential.”

We appreciate the reviewer’s valuable comment. In the revised manuscript, we have conducted LSV measurement in 0.4 M KNO₃ to investigate the OER competition over carbon paper. As shown in Figure R1, the current density in 0.4 M KNO₃ was far lower than that in 0.4 M KBr within the whole potential range. The potentials at 10 mA cm⁻² in 0.4 M KBr and 0.4 M KNO₃ were 0.9 and 1.8 V vs Ag/AgCl, respectively. Even at 4.9 V vs Ag/AgCl, the current density in 0.4 M KBr was 1.7 times higher than that in 0.4 M KNO₃. Moreover, as depicted in Supplementary Figure 17a, only trace amount of O₂ was detected via GC at all applied potentials. These results sufficiently confirmed that the electrooxidation of Br⁻ was still superior to OER even at high applied potentials over carbon paper.

To clarify the intrinsic reason for the inhibited OER, we have calculated the electrode potential for the Br⁻ electrooxidation in our system.

$$E = E^0 + (RT / nF) \times \ln [\text{Br}_2 / (\text{Br}^-)^2] = 1.08 + 0.0592 / 2 \times \lg [\text{Br}_2 / (0.4)^2]$$

The standard electrode potential depended on the concentration of Br⁻ and Br₂. Since the generated Br₂ was continuously pumped out of the cell in the decoupled system, the local concentration of Br₂ in the electrolytic cell was extremely low. In this case, the actual electrode potential of the Br⁻ electrooxidation was more negative than the standard electrode potential of 1.08 V vs RHE. Second, OER was kinetically unfavorable relative to Br⁻ electrooxidation due to the different involved electron numbers in the two reactions (four-electron reaction for OER vs

two-electron reaction for Br⁻ electrooxidation). As such, the overpotential of OER was commonly higher than that of Br⁻ electrooxidation [*Adv. Funct. Mater.* **31**, 2101820 (2021)]. Third, the overpotential of Br⁻ electrooxidation was significantly different over various electrodes. We have measured LSV in 0.4 KBr using a variety of commercial materials as anodes including Pt foil, dimensionally stable anode (DSA), stainless steel foil, and titanium suboxide (Ti₄O₇) electrodes (Supplementary Fig. 2). Notably, the overpotential of the carbon paper was lower than other commercial electrodes, suggesting that the carbon paper exhibited excellent performance for Br⁻ electrooxidation. To sum up, the inhibition of OER could be ascribed to the decreased standard electrode potential of the Br⁻ electrooxidation, the sluggish kinetics of OER, and the low overpotential of Br⁻ electrooxidation over the carbon paper.

“In addition, KBr concentration is generally sensitive to bromine evolution reaction. Why do authors select 0.4 M KBr as electrolyte.”

We genuinely thank this reviewer for pointing out the issue. Actually, we initially tested the catalytic performance in different concentrations of Br⁻, which suggested that the concentration of 0.4 M was optimal. As shown in Supplementary Figure 7, the FE_{PO} was improved from 83% to 91% as the concentration of KBr was increased from 0.2 to 0.4 M. The enhanced FE_{PO} could be ascribed to the facilitated mass diffusion of Br⁻ at high concentrations. When the concentration of electrolyte was further increased to 5 M, the FE_{PO} gradually decreased to 32%. This result validated that excessive Br⁻ was detrimental to the reversible disproportionation reaction of Br₂ with water ($\text{Br}_2 + \text{H}_2\text{O} \rightleftharpoons \text{HBrO} + \text{HBr}$), thus inhibiting the formation of HBrO. As such, the concentration of KBr in our work was set to 0.4 M to evaluate the performance of PO production. We have added the corresponding discussions in the revised manuscript (lines 21 to 30 on page 5, highlighted in yellow color).

“6. Is there possibility that the improved FE compared with the chlorohydrin route is related to the inhibited HBrO dissociation? Is there any solubility issue associated with propylene when it reacts with HBrO?”

We are thankful for the issue raised by the reviewer. In the revised manuscript, we have conducted iodometric titration experiments to determine the amount of BrO⁻ in the anolyte after the reaction. As shown in Supplementary Figure 9b, the FE for BrO⁻ was 3.9% at 1.9 V vs Ag/AgCl, which was lower than that (25%) of the chlorohydrin route in the previous report [*Science* **368**, 1228-1233 (2020)]. Hence, the improved FE compared with the chlorohydrin route could be partly related to the inhibited HBrO dissociation.

In this work, the spontaneous reaction between propylene and HBrO could react rapidly once contacted. In addition, the active HBrO would not become invalid for a time, which provided enough time for the propylene conversion. Therefore, we think that the solubility of propylene would not significantly affect the reaction performance. In the previous manuscript, to increase the contact interface between propylene and HBrO, we dispersed the propylene into dense bubbles by utilizing a sand core airway. Combined with the vigorous stir, the collision probability between propylene and HBrO was further increased, thereby ensuring the reaction efficiency. The corresponding experimental details have been added in the **Electrochemical measurements** section.

“7. It is suggested to verify the conversion and yield rates of the chemical steps involved in the formation of C₃H₇BrO and PO. This verification could be achieved, for example, by introducing HBrO solution to confirm the production of PO.”

We genuinely thank this reviewer for his/her constructive suggestion. As suggested, according to the amount of PO generated at 1.9 V vs Ag/AgCl, we bubbled propylene into a comparable amount of Br₂ water at the same flow rate in our work, without applying any potentials. As shown in Figure R2a, the propylene could also be converted into C₃H₇BrO and PO. In addition, the yield rates for C₃H₇BrO and PO were equivalent to the values in our work, indicating that the propylene conversion and saponification processes were spontaneous (Figure R2b).

“8. The FE of generated H₂ at the cathode and the catholyte/anolyte pH levels were not mentioned in the manuscript. After the Br₂ disproportionation reaction, it is expected that the anolyte could be strongly acidic so may need substantial OH⁻ from the cathode to neutralize it. Thus, it is important to determine the OH⁻ produced in the cathode where the catholyte pH could be an important factor to limit the kinetics of PO formation step.”

We appreciate the reviewer’s suggestion. As reminded by this reviewer, we have determined the FE for H₂ at the cathode via GC. As shown in Supplementary Figure 11, the FE for H₂ was nearly 100% at all applied potentials (lines 17 to 18 on page 6, highlighted in yellow color).

In the revised manuscript, we have determined the pH of the anolyte and catholyte after the reaction via a pH meter. The pH of the anolyte and catholyte operated at 1.9 V vs Ag/AgCl were 2.16 and 12.17, respectively. According to the pH, the concentrations of H⁺ and OH⁻ were calculated to be 6.9 and 14.8 mmol L⁻¹, respectively. The concentration of generated C₃H₇OBr was determined to be 7.0 mmol L⁻¹. As such, the OH⁻ generated at the cathode was adequate to neutralize the anolyte. Furthermore, we have also prepared 1 M KOH with a pH of 14.00 to explore the effect of catholyte pH. However, more than half of PO was hydrolyzed into 1,2-propylene glycol, indicating that excess OH⁻ restricted the selectivity for PO (Figure R3).

“9. In TEA model, reasonable references should be added. The electrolyser costs and prices of feedstock and products should re-calculate and re-set according to current market reports, respectively. Why parameters of model (full cell voltage, current density, etc.) are not based on experimental results? TEA results should be re-calculated.”

We sincerely thank this reviewer for his/her advice. We have re-set the prices of feedstocks, products, and electrolyzers and re-calculated the TEA in the revised manuscript. The price data of feedstocks and products originated from <https://www.chemanalyst.com>. The cost of the electrolyzer was re-calculated according to recent reports [*J. Am. Chem. Soc.* **146**, 714-722 (2024); *Nat. Sustain.* **6**, 827-837 (2023); *Nat. Sustain.* **4**, 911-919 (2021)]. Notably, the trend of data was similar to the previous result.

Besides, all parameters in the previous manuscript were indeed based on the experimental results including applied potentials, current densities, FE, and so on. However, we calculated the TEA based on the half-cell applied potentials. According to the reported literature, adopting the cell

voltage to calculate the TEA was more reasonable. As reminded by this reviewer, we have revised this parameter and supplemented more calculation details in the **Techno-economic analysis** section.

Reviewer #2:

“The manuscript by Zeng et al. reported an electrochemical bromohydrin route for highly efficient synthesis of propylene oxide. The strategy integrated electrolysis and propylene by coupling electrolysis of bromide and propylene conversion process within separated reactors. The system demonstrates the capability for continuous propylene oxide production over a month at high current density, maintaining both high selectivity and Faradaic efficiency. While the work is intriguing and showcases the potential for large-scale epoxide production to replace traditional methods, the manuscript could benefit from the inclusion if there are more fundamental and mechanistic data. Strengthening the manuscript with additional principal data would better align with the requirements of a scientific paper, rather than solely focusing on the technological process and its results, if the manuscript will be published in this journal. Nevertheless, the manuscript could potentially benefit from the following suggestions.”

We sincerely thank this reviewer for his/her careful reading of our manuscript. According to the reviewer’s constructive comments, we have supplemented Tafel and reaction order analyses to investigate the mechanism of Br^- electrolysis over the carbon paper. In addition, we have also conducted in situ Raman experiments to further verify the reaction mechanism of the whole process. We hope the revised version could address the concerns raised by this reviewer.

According to the Tafel analysis, we verified that Br_2 was formed via the Volmer-Heyrovsky mechanism. Both Tafel and reaction order results demonstrated the rate-determining step (RDS) was the Heyrovsky step (Supplementary Fig. 4). To monitor the whole process, we have conducted in situ Raman experiments in the revised manuscript. When the potential was applied, a discernible peak at 309 cm^{-1} was observed, corresponding to the Br-Br bond in Br_2 (Supplementary Fig. 1a). When the applied potential increased from 1.9 to 2.9 V vs Ag/AgCl, the peak intensity of Br_2 was gradually enhanced, indicating the growing concentration of Br_2 in the electrolyte. We have also measured the Raman spectra of the products after the addition reaction between HBrO with propylene and the saponification reaction between the anolyte and catholyte. As shown in Supplementary Figure 1b, after propylene was bubbled into the anolyte, the peaks at 664 and 826 cm^{-1} were ascribed to the C-Br and C-C bonds of $\text{C}_3\text{H}_7\text{OBr}$, respectively. In addition, after the saponification reaction by mixing the anolyte and catholyte, the peaks at 952 and 1274 cm^{-1} corresponding to the C-O-C of PO were observed (Supplementary Fig. 1c). The generation of $\text{C}_3\text{H}_7\text{OBr}$ and PO was further determined via ^1H NMR. These results collectively provided supporting evidence for the proposed reaction mechanism (lines 1 to 4 and lines 16 to 18 on page 5, highlighted in yellow color).

“1. Whether the authors have investigated other commercial electrode materials besides carbon paper. The detail reaction mechanism involved with Br^- on carbon paper would help to improve understanding and thoughtful of the manuscript.”

We sincerely thank this reviewer for his/her valuable suggestion. Actually, we initially investigated the performance of various commercial electrodes including Pt foil, dimensionally stable anode (DSA), stainless steel foil, and titanium suboxide (Ti_4O_7) electrodes via LSV measurements in 0.4 M KBr electrolyte. As depicted in Supplementary Figure 2, the carbon

paper exhibited a higher current density than other commercial electrodes, implying the superior catalytic performance of the carbon paper. As such, the carbon paper was chosen as the anode.

To investigate the reaction mechanism involved with Br⁻ electrooxidation over the carbon paper, we have conducted the Tafel and electrochemical reaction order analyses in the revised manuscript. The Br⁻ electrooxidation consists of three reaction mechanisms as follows:

Reaction mechanism	Reaction equation	Tafel slope (mV dec ⁻¹)
Volmer step	$\text{Br}^- \rightarrow \text{Br}_{\text{ads}} + \text{e}^-$	120
Heyrovsky step	$\text{Br}^- + \text{Br}_{\text{ads}} \rightarrow \text{Br}_2 + \text{e}^-$	40
Volmer step	$\text{Br}^- \rightarrow \text{Br}_{\text{ads}} + \text{e}^-$	120
Tafel step	$2\text{Br}_{\text{ads}} \rightarrow \text{Br}_2$	30
Krishtalik step	$\text{Br}^- \rightleftharpoons \text{Br}_{\text{ads}} + \text{e}^-$ $\text{Br}_{\text{ads}} \rightarrow \text{Br}_{\text{ads}}^+ + \text{e}^-$ $\text{Br}_{\text{ads}}^+ + \text{Br}^- \rightleftharpoons \text{Br}_2$	/

The Tafel slopes were 120, 40, and 30 mV dec⁻¹ with the rate-determining steps of Volmer, Heyrovsky, and Tafel steps, respectively [*Sci Rep* **5**, 13801 (2015)]. As for the Krishtalik step, it was verified only occurred over metal oxides because the electron-rich oxide lattice could stabilize the Br_{ads}⁺ species [*J. Electrochem. Sci. Technol.* **14**, 105-119 (2023)]. As shown in Supplementary Figure 4a, the Tafel plot of the carbon paper was 55 mV dec⁻¹, demonstrating that Br₂ was formed via the Volmer-Heyrovsky mechanism with the Heyrovsky step as the rate-determining step. Moreover, the low Tafel slope indicated the fast reaction kinetics of the carbon paper. Additionally, we also evaluated the reaction order dependence behavior in different concentrations of KBr electrolytes. As shown in Supplementary Figure 4b, the reaction order was fitted to be 0.89, suggesting a roughly first-order dependence on Br⁻ concentration. Since the Heyrovsky step resembled the Eley-Rideal-type desorption involving only one Br⁻ adsorption, it could be concluded that the rate-determining step of Br⁻ electrooxidation over the carbon paper was the Heyrovsky step, which was consistent with the Tafel results [*Electrochim. Acta* **37**, 51-63 (1992); *J. Electrochem. Sci. Technol.* **14**, 105-119 (2023)]. We have added the corresponding discussions in the revised manuscript (lines 9 to 12 and lines 16 to 18 on page 5, highlighted in yellow color).

“2. It is essential to provide the utilization rate of substrate in this reaction system, particularly for gas substrate, considering the high cost.”

We genuinely thank this reviewer for his/her constructive suggestion. Actually, we determined the conversion of propylene in the previous manuscript. Notably, the single-pass conversion of propylene achieved 66% at 6.25 A (Supplementary Fig. 22). Moreover, since the selectivity of propylene conversion was above 99.9%, the unreacted propylene could be collected and recycled directly. Hence, the propylene was expected to be completely converted after multiple cycles.

“3. To reflect the obvious advantages of the electrochemical bromohydrin route, the author could make a cost-comparison with traditional industrial process, such as H₂O₂ and O₂.”

We sincerely thank this reviewer for his/her comments. As suggested, we have compared our work with traditional industrial processes using H₂O₂ and O₂ as additional oxidants. The prices of propylene and PO in 2023 were ~1200 and ~1700 \$/ton, respectively (price data all originated from <https://www.chemanalyst.com>). According to the TEA results, the profit and total cost of our work were 231 and 1502 \$/ton, respectively (Fig. 3a). For the HPPO method, the profit was calculated to be 795 \$/ton according to the reported literature [*ACS Sustainable Chem. Eng.* **1**, 268-277 (2013)]. However, the prices of propylene, PO, and H₂O₂ in that work were assumed to be 1210, 2670, and 529 \$/ton. With the declined price of PO (~1700 \$/ton) and the soaring price of H₂O₂ (~700 \$/ton), the price difference between propylene and PO was not enough to offset the additional H₂O₂, causing this process economically unfavorable [*ACS Catal.* **10**, 13415-13436 (2020)]. As for the direct oxidation method, although the price of O₂ was only ~20 \$/ton, the harsh operating conditions resulted in the high cost. According to the detailed report, the total cost was ~7255 \$/ton, which was higher than that of our work [*Nat Commun.* **13**, 7504 (2022)]. With the development of sustainable energy, the price of electricity has been significantly reduced, making the electrochemical conversion profitable. Hence, the electrochemical bromohydrin route in our work was cost-effective.

“4. The assertion of “breaking the limit of the current” is overstatement due to the decreased energy efficiency at elevated working potentials. What does the high temperature of reactors resulting from high cell voltage impact the reaction?”

We sincerely thank this reviewer for raising this issue. In fact, the current in previous works was limited. Specifically, the alkenes were bubbled or placed into the electrolyzer directly in previous works. In this case, the alkenes inevitably underwent unwanted side reactions at high currents such as overoxidation, which was not conducive to PO production at high currents, thus restricting the scalable production of PO. In our work, the conversion of propylene would not be affected by the applied current since the propylene conversion process and the electrolysis process were separated. In this case, the performance could be maintained even at high currents. To avoid the misconception to reviewers, we have revised the “breaking the limit of the current” to “which was expected to maintain the performance even at high currents” in the revised manuscript. Besides, the energy efficiency for PO mainly depended on the applied potential and FE_{PO}. Since the FE_{PO} remained stable even at high applied potentials in our work, it was reasonable that the energy efficiency decreased at high applied potentials.

To evaluate the impact of the temperature, we have measured the performance at different temperatures in the revised manuscript, including 20, 30, and 40 °C. As shown in Supplementary Table 1, FE_{PO} was maintained stable within the whole temperature range. At the same time, the j_{PO} was gradually enhanced as the temperature increased. This phenomenon could be attributed to the lower electrolyzer impedance and the accelerated mass transfer of Br⁻ at higher temperatures [*EES Catal.* **1**, 934-949 (2023)]. We have also measured the electrolyte temperature operated at 4.9 V vs Ag/AgCl, which was determined to be 29 °C. Based on the above results, we think this proper high temperature might be conducive to the improvement of the current. However, it was undeniable that the device (including the cell, membrane, and so on) might be

damaged when the temperature rose to a certain level [*EES Catal.* **1**, 934-949 (2023)]. At the same time, the side reaction might be more severe at higher temperatures. As such, considering the temperature accumulation during the long-term test, the electrolyte tank was maintained at 20 °C controlled by a cold trap when evaluating the stability. In this case, the system could run stably for 750 h. We have added the corresponding discussions in the revised manuscript (lines 21 to 26 on page 6, highlighted in yellow color).

Reviewer #3:

In this work, the authors developed an impressive strategy by spatially decoupling the electrolysis process and the propylene conversion process for efficient propylene oxide (PO) production. This strategy achieved record-high FE (91%), product selectivity (100%), and long-term stability (operation for >30 days) relative to other electrochemical processes. In addition, the authors designed an enlarged flow reactor with a geometric electrode area of 25 cm² to scale up the manufacture of PO. Combined with the designed separation device, the practical application promise of this strategy was demonstrated. Finally, this strategy could also be applied to the efficient transformation of other alkenes, including gaseous and liquid alkenes. This work is impressive and well-organized in general. Therefore, I recommend the publication of this manuscript on Nature Communications after addressing a few minor questions.

We sincerely thank this reviewer for his/her careful reading of our manuscript and positive appraisal.

“-In this strategy, the concentration of bromine ions may affect the reaction efficiency. It is necessary to provide the experimental results for optimizing the concentration of bromine ions.”

We genuinely thank this reviewer for his/her constructive suggestion. Actually, we initially tested the catalytic performance in different concentrations of Br⁻, which suggested that the concentration of 0.4 M was optimal. As shown in Supplementary Figure 7, the FE_{PO} was improved from 83% to 91% as the concentration of KBr was increased from 0.2 to 0.4 M. The enhanced FE_{PO} could be ascribed to the facilitated mass diffusion of Br⁻ at high concentrations. When the concentration of electrolyte was further increased to 5 M, the FE_{PO} gradually decreased to 32%. This result validated that excessive Br⁻ was detrimental to the reversible disproportionation reaction of Br₂ with water ($\text{Br}_2 + \text{H}_2\text{O} \rightleftharpoons \text{HBrO} + \text{HBr}$), thus inhibiting the formation of HBrO. As such, the concentration of KBr was set to 0.4 M to evaluate the performance of the electrochemical transformation of propylene into PO. We have added the corresponding discussions in the revised manuscript (lines 21 to 30 on page 5, highlighted in yellow color).

“-Bromine may also be further oxidized to other high valence products such as bromic acid or added to organic substrates to produce brominated products. The formation of these by-products should be excluded.”

We sincerely thank this reviewer for raising this constructive comment. In the revised manuscript, we have determined the anion in the anolyte after electrolytic reaction via ion chromatography. As shown in Supplementary Figure 9a, the BrO₃⁻ and other high-valence bromine ions were below the detection limit.

In our previous manuscript, we explored the possibility of Br₂ engaging in an addition reaction with propylene to produce brominated products such as 1,2-dibromopropane (C₃H₆Br₂). As determined by ¹H NMR, no byproducts were formed after the addition reaction (Supplementary Fig. 5a-b). We have added the corresponding discussions in the revised manuscript (lines 6 to 9 and lines 12 to 15 on page 6, highlighted in yellow color).

“-The FE for H₂ in the whole reaction process was not measured, so it was not rigorous to directly assume the 100%-FE for H₂ when calculating techno-economic analysis (TEA). Therefore, the authors are advised to determine the FE for H₂ in the whole reaction process to calculate the TEA.”

We sincerely thank this reviewer for his/her advice. Actually, the cathode only underwent the hydrogen evolution reaction (HER) during electrolysis. As reminded by this reviewer, we have determined the FE for H₂ at the cathode via GC. As shown in Supplementary Figure 11, the FE for H₂ was nearly 100% at all applied potentials. Therefore, we calculated the TEA with the FE for H₂ of 100% (lines 17 to 18 on page 6, highlighted in yellow color).

“-In the TEA section, the assumption on the price of electricity “which is higher than the price of industrial electricity” was presented without sufficient supporting reference. More references should be listed to support the assumption.”

We genuinely thank this reviewer for raising the comments. The electricity price mentioned in the manuscript was obtained by referring to some related literature [ACS Energy Lett. **6**, 997-1002 (2021); Energy Environ. Sci. **11**, 1536-1551 (2018); Ind. Eng. Chem. Rec. **57**, 2165-2177 (2018)]. As suggested, we have supplemented more literature to support our hypothesis [Nat. Catal. **3**, 14-22 (2022); Nat. Sustain. **6**, 827-837 (2023)]. Our assumption of 15 cents/kWh for the electricity price was far higher than that in these reports. Moreover, we have referred to the latest literature and conducted the TEA with the electricity price of 10 cents/kWh in the revised manuscript [Nat. Sustain. **6**, 827-837 (2023); J. Am. Chem. Soc. **146**, 714-722 (2024)].

“-The post-reaction structure of the carbon paper was not well characterized. More characterizations of the spent catalyst after the durability test should be offered.”

We genuinely thank this reviewer for his/her comments. In the revised manuscript, we have conducted X-ray photoelectron spectroscopy (XPS) characterization to demonstrate the structural information of the carbon paper before and after the stability reaction. For the pristine carbon paper, C 1s and O 1s peaks were observed in the survey spectra (Supplementary Fig. 14a). Especially, the C 1s region of the pristine carbon paper was deconvoluted into four peaks at 284.5, 284.8, 285.6, and 286.5 eV, corresponding to C=C, C-C, C-O, and C=O bonds, respectively (Supplementary Fig. 14b) [Nat. Commun. **9**, 3376 (2018); Sci Rep **10**, 6902 (2020)]. As for the carbon paper after the reaction, apart from the peaks of C 1s and O 1s, two new peaks ascribed to Br 3d and 3p were observed (Supplementary Fig. 14d). The C 1s region was deconvoluted into five peaks at 284.5, 284.8, 285.6, 286.1, and 286.5 eV, respectively. The new peak located at 286.1 eV was assigned to the C-Br bonds (Supplementary Fig. 14e) [Fuller. Nanotub. Carbon Nanostruct. **28**, 1048-1058]. The formation of C-Br bonds was further confirmed by the Br 3d spectrum. As shown in Supplementary Figure 14f, the two peaks at 70.1 and 71.2 eV corresponded to the C-Br bonds [Sci Rep **10**, 6902 (2020)]. It has been reported that the formed C-Br bonds were considered to blanket the surface of the carbon paper from being further attacked, thus contributing to the ultra-long stability (>30 days) for the carbon paper in our work [Carbon **29**, 165-171 (1991); Electrochim. Acta **56**, 2246-2253 (2011)]. Interestingly, the contents of O and Br in the carbon paper increased initially and remained relatively stable

due to the protection of C-Br bonds (Supplementary Table 2). Hence, we conclude that the functional groups of C-Br on the carbon paper surface might protect the bulk from further oxidation during the electrolysis process. In the revised manuscript, we have added the corresponding discussions (lines 5 to 10 on page 7, highlighted in yellow color).

“-The experimental evidence about the reaction mechanism was limited. I suggest the authors supplement relevant experiments to verify the reaction mechanism.”

We genuinely thank this reviewer for his/her valuable suggestion. In the revised manuscript, we have carried out Tafel and reaction order analyses to investigate the mechanism of Br⁻ electrolysis over carbon paper. The Br⁻ electrooxidation consists of three reaction mechanisms as follows:

Reaction mechanism	Reaction equation	Tafel slope (mV dec ⁻¹)
Volmer step	$\text{Br}^- \rightarrow \text{Br}_{\text{ads}} + \text{e}^-$	120
Heyrovsky step	$\text{Br}^- + \text{Br}_{\text{ads}} \rightarrow \text{Br}_2 + \text{e}^-$	40
Volmer step	$\text{Br}^- \rightarrow \text{Br}_{\text{ads}} + \text{e}^-$	120
Tafel step	$2\text{Br}_{\text{ads}} \rightarrow \text{Br}_2$	30
Krishtalik step	$\text{Br}^- \rightleftharpoons \text{Br}_{\text{ads}} + \text{e}^-$ $\text{Br}_{\text{ads}} \rightarrow \text{Br}_{\text{ads}}^+ + \text{e}^-$ $\text{Br}_{\text{ads}}^+ + \text{Br}^- \rightleftharpoons \text{Br}_2$	/

The Tafel slopes were 120, 40, and 30 mV dec⁻¹ with the rate-determining steps of Volmer, Heyrovsky, and Tafel steps, respectively [*Sci Rep* **5**, 13801 (2015)]. As for the Krishtalik step, it was verified only occurred over metal oxides because the electron-rich oxide lattice could stabilize the Br_{ads}⁺ species [*J. Electrochem. Sci. Technol.* **14**, 105-119 (2023)]. As shown in Supplementary Figure 4a, the Tafel plot of the carbon paper was 55 mV dec⁻¹, demonstrating that Br₂ was formed via the Volmer-Heyrovsky mechanism with the Heyrovsky step as the rate-determining step. Moreover, the low Tafel slope indicated the fast reaction kinetics of the carbon paper. Additionally, we also evaluated the reaction order dependence behavior in different concentrations of KBr electrolytes. As shown in Supplementary Figure 4b, the reaction order was fitted to be 0.89, suggesting a roughly first-order dependence on Br⁻ concentration. Since the Heyrovsky step resembled the Eley-Rideal-type desorption involving only one Br⁻ adsorption, it could be concluded that the rate-determining step of Br⁻ electrooxidation over the carbon paper was the Heyrovsky step, which was consistent with the Tafel results [*Electrochim. Acta* **37**, 51-63 (1992); *J. Electrochem. Sci. Technol.* **14**, 105-119 (2023)].

Moreover, we have conducted in situ Raman experiments to monitor the whole process. When the potential was applied, a discernible peak at 309 cm⁻¹ was observed, corresponding to the Br-Br bond in Br₂ (Supplementary Fig. 1a). When the applied potential increased from 1.9 to 2.9 V vs Ag/AgCl, the peak intensity of Br₂ was gradually enhanced, indicating the growing concentration of Br₂ in the electrolyte. We have also measured the Raman spectra of the products

after the addition reaction between HBrO with propylene and the saponification reaction between the anolyte and catholyte. As shown in Supplementary Figure 1b, after propylene was bubbled into the anolyte, the peaks at 664 and 826 cm^{-1} were ascribed to the C-Br and C-C bonds of $\text{C}_3\text{H}_7\text{OBr}$, respectively. In addition, after the saponification reaction by mixing the anolyte and catholyte, the peaks at 952 and 1274 cm^{-1} corresponding to the C-O-C of PO were observed (Supplementary Fig. 1c). The generation of $\text{C}_3\text{H}_7\text{OBr}$ and PO was further determined via ^1H NMR. These results collectively provided supporting evidence for the proposed reaction mechanism. We have added the corresponding discussions in the revised manuscript (lines 1 to 4 on page 5 and lines 16 to 18 on page 5, highlighted in yellow color).

“-The discussion about Faradaic efficiency was presented without much context or explanation. In-depth discussion and explanation of this result is necessary.”

We appreciate the reviewer’s constructive suggestions. In the revised manuscript, we have added the corresponding discussions and explanations for Faradaic efficiency. The details are supplied as follows.

Figure 2b shows the FE and the selectivity for PO. It is worth noting that the selectivity for PO reached above 99.9% at all applied potentials. No brominated products were detected (Supplementary Fig. 5a-b). This result could be attributed to the decoupled system, which avoided the direct contact between propylene and the high-concentration Br_2 at the anode surface. The FE_{PO} was maintained above 88% at all applied potentials. Especially, when the applied potential was set to 1.9 V vs Ag/AgCl, the highest FE_{PO} of 91% was achieved. In addition, the FE for $\text{C}_3\text{H}_7\text{OBr}$ was close to FE_{PO} at the corresponding applied potentials (Supplementary Fig. 9). To demonstrate whether high-valance bromine ions were produced, we tested the anion in the anolyte after electrolytic reaction via ion chromatography. As shown in Supplementary Figure 10a, the BrO_3^- or other high-valance bromine ions in the anolyte were below the detection limit. Besides, the FE for BrO^- was 3.9% at 1.9 V vs Ag/AgCl determined by iodometric titration experiments (Supplementary Fig. 10b). The FE for H_2 generated at the cathode was quantified via online GC, which corresponded to nearly 100% at all applied potentials (Supplementary Fig. 11). We also measured the performance at different temperatures including 20, 30, and 40 $^\circ\text{C}$ to evaluate the temperature impact at high applied potentials. Interestingly, FE_{PO} was maintained stable within the whole temperature range. At the same time, the j_{PO} was gradually enhanced as the temperature increased (Supplementary Table 1). This phenomenon could be attributed to the lower electrolyzer impedance and the accelerated mass transfer of Br^- at higher temperatures. In the revised manuscript, we have added the corresponding discussions (lines 5 to 18 and lines 21 to 26 on page 6, highlighted in yellow color).

“-Some figure captions are unclear, such as the Supplementary Figure 2 and Supplementary Figure 5. The authors should check and improve the content of the figure captions carefully.”

We sincerely thank this reviewer for pointing out this issue. We have revised the captions of Supplementary Figure 2 and Supplementary Figure 5 to strengthen the connection between the contents of the pictures and the text of the manuscript. We have also rechecked and modified all the figures and figure captions in the revised manuscript.

“-Some experimental details should be supplemented. For example, for the H-cell experiments, the volume of the cell was not mentioned. A thorough check and proofread is necessary.”

We genuinely thank this reviewer for pointing out this valuable suggestion. In the revised manuscript, we have added the relevant information in the **Electrochemical measurements** section including the volume of the cell, the flow rate of propylene, and so on.

REVIEWERS' COMMENTS

Reviewer #1 (Remarks to the Author):

The authors has successfully addressed my concerns. The manuscript is suitable for publishing.

Reviewer #2 (Remarks to the Author):

The authors have made a good revision. I suggest the publication of this revised version as such.

Reviewer #3 (Remarks to the Author):

The authors have addressed all my comments and the manuscript can be accepted now.

Point-by-point response to reviewers' comments

Manuscript ID: NCOMMS-23-51936A

MS Type: Article

Title: "Linking propylene and anode via bromide mediator in separated reactors for efficient propylene oxide synthesis"

Reviewer #1 (Remarks to the Author):

"The authors has successfully addressed my concerns. The manuscript is suitable for publishing."

We sincerely thank this reviewer for his/her suggestions which helped us improve this work during revision.

Reviewer #2 (Remarks to the Author):

"The authors have made a good revision. I suggest the publication of this revised version as such."

We sincerely thank this reviewer for his/her approval and help on the improvement of this work.

Reviewer #3 (Remarks to the Author):

"The authors have addressed all my comments and the manuscript can be accepted now."

We sincerely thank the reviewer for his/her positive comments.